# The Human Aspect of Horse Care: How the COVID-19 Pandemic Impacted the Wellbeing of Equestrian Industry Stakeholders

**DOI:** 10.3390/ani11082163

**Published:** 2021-07-22

**Authors:** Ashley Ward, Kate Stephen, Caroline Argo, Christine Watson, Patricia Harris, Madalina Neacsu, Wendy Russell, Dai Grove-White, Philippa Morrison

**Affiliations:** 1Scotland’s Rural College, Aberdeen AB21 9YA, UK; Ashley.ward@sruc.ac.uk (A.W.); Kate.Stephen@sruc.ac.uk (K.S.); caroline.argo@sruc.ac.uk (C.A.); christine.watson@sruc.ac.uk (C.W.); 2The Rowett Institute, University of Aberdeen, Foresterhill, Aberdeen AB25 2ZD, UK; m.neacsu@abdn.ac.uk (M.N.); W.russell@abdn.ac.uk (W.R.); 3Equine Studies Group, WALTHAM Petcare Science Institute, Leicestershire LE14 4RT, UK; pat.harris@effem.com; 4Department of Epidemiology & Population Health, School of Veterinary Science, Faculty of Health and Life Sciences, University of Liverpool, Liverpool L69 3BX, UK; daigw@liverpool.ac.uk

**Keywords:** pandemic, mental health, equine, qualitative, interviews, welfare

## Abstract

**Simple Summary:**

The COVID-19 pandemic resulted in significant changes in the way people live and work. To better inform responses to any future pandemics, it is important for researchers to investigate the ways in which different groups were impacted. Those involved in the care of horses are one such group who faced a variety of challenges to their own wellbeing whilst they navigated providing care to animals during the UK’s initial lockdown. The present research utilised a qualitative approach to investigate the impacts of the pandemic on the wellbeing of groups of horse owners, equine veterinarians, farriers and welfare centre managers, mostly based in the Aberdeenshire region. Results from this study showed that changes in communication style and restricting people’s access to their horses had negative mental health consequences for veterinarians and horse owners, respectively. It also identified that this equestrian community sample had developed ways to improve their own wellbeing through pro-social attitudes and activities which also benefited the wider community. It is hoped that these findings will help to protect and promote wellbeing in future pandemic scenarios.

**Abstract:**

During the lockdown phase of the COVID-19 pandemic, equestrian stakeholders faced a dilemma whereby they were required to balance caring for the welfare of horses with adapting to the restrictions imposed to protect public health. The present study investigated the impact of the pandemic on the wellbeing of a sample of industry stakeholders, including horse owners, equine veterinarians, farriers and welfare centre managers (*n* = 26) using a qualitative methodology. Findings from the interviews indicated that the mental health and wellbeing of veterinarians and horse owners was negatively affected by pandemic-related obstacles to communication and limitations to horse–owner interactions. However, this study also identified several positive outcomes for wellbeing during lockdown resulting from pro-social activities that were engaged with by horse owners to overcome social isolation, the separation of the community and loneliness. These findings provide accounts of ways in which those caring for horses might be challenged during national emergency scenarios, pointing to areas that would benefit from future mental health and wellbeing interventions.

## 1. Introduction

The COVID-19 pandemic has had wide-reaching consequences for public health on a global scale. Research efforts to quantify and address the impact of the pandemic on mental health and wellbeing have registered a decline in measures of wellbeing in human populations across the globe [1,2,3,4,5].

Challenges for societal mental health and wellbeing during the pandemic include coping with contagion, isolation and financial instability [6]. Factors associated with mental health outcomes, such as loneliness, stress and unemployment, likely evolved as the nation progressed through the stages of lockdown. During the slow release then re-enforcement of restrictions, and then the gradual move towards “normality” [7], individuals were required to rapidly adapt to the changing situation, with access to amenities and freedom of movement being dictated by locality, profession and individual circumstances. As an industry, equestrianism is often thought to host attitudes of resilience and bravado, with little room for fear or anxiety from horse owners, particularly riders [8]. However, the COVID-19 pandemic was an indiscriminatory source of distress for all communities, including equestrians. Focused evaluations on the impact of the pandemic upon distinct social cohorts can offer insights into the ways that social isolation, contagion and restricted living interact with sense of overall wellbeing within a community. Equestrian industry stakeholders form one such group which has received considerable attention from researchers seeking to understand the full impact of the pandemic upon human and animal welfare [9,10,11,12,13,14,15].

Owner wellbeing has been linked to the welfare of the horse [13], with a healthy, comfortable horse providing a benchmark for the successful fulfilment of the role for some horse owners. However, horse welfare involves a network of support, which includes, in particular, input from an owner, veterinarian, farrier and sometimes a professional care provider. When the network formed by these individuals is broken, for example if an owner can no longer afford for a veterinarian or farrier to attend their horse, the animal’s welfare can be compromised and there is a potential negative impact on the wellbeing of the owner and others in the equine care team. To evaluate the impact of the pandemic on this primary team of equine care providers, a qualitative approach was utilised.

Semi-structured interviews were conducted with equestrian industry stakeholders, including equine veterinarians, farriers, horse owners and equine welfare centre managers, with the aim of documenting the experience of this group in response to the first lockdown phase of the pandemic. Amongst other topics broadly relating to the management of native ponies and associated challenges during the pandemic, interviewees discussed the stressors associated with changing their means of communication whilst adapting to restricted lifestyles and work conditions.

## 2. Materials and Methods

The qualitative approach can be found in more detail elsewhere [9]. This study was designed within a grounded theory methodological framework, encouraging the identification of themes in early interviews that would guide the focus of this study and the proceeding discussions based on the most important topics highlighted by stakeholders themselves. Ethical approval from the SRUC Social Science Ethics Committee was obtained on 1 May 2020, and all interviewees provided written (online) and verbal (recorded) informed consent to take part in this study.

A purposeful, pragmatic sampling approach was used to secure a heterogenous sample, resulting in 24 interviews with equine or mixed-practice (with an interest in equine) veterinarians (*n* = 5), horse owners with horses at home (*n* = 6), owners with horses kept at an external establishment, known as livery yards (*n* = 5), farriers (*n* = 4) and equine welfare centre managers (*n* = 4). All interviewees were based in the Aberdeenshire region of Scotland except two equine welfare centre managers (based in England).

Pre-defined questions were used to direct semi-structured interviews broadly relating to the management of native ponies and associated challenges during the pandemic. However, conversation with interviewees frequently drifted towards areas of their own particular interest which included, or signalled, mental health and wellbeing. Although mental health was not a key focus during study design, its importance in interviewee conversations led the researchers to dedicate an analysis to the ways in which mental health and wellbeing were impacted by the pandemic.

Interviews lasted an average of 32 min, and audio recordings of the interviews were transcribed verbatim and anonymised prior to secure storage. Transcripts were thereafter referred to by their category and randomised order of interview (e.g., horse owner with horse at home 1 (HH1), horse owner with horse at livery 2 (HL2), equine veterinarian 3 (V3), farrier 4 (F4) and equine welfare centre manager 5 (WCM5)). Interview transcripts were analysed in NVivo Windows (1.0) using thematic analysis with an approach rooted grounded theory.

Initially, first-order coding was performed to understand and contextualise the raw data set. Inductive codes evolved during the interview period between May and July of 2020, allowing the exploration of patterns and themes as this study progressed. Codes were then refined into broad subthemes and themes which captured the concepts perceived to be the most prominent across the data set, giving rise to two subject areas, “Policy Implications of the Pandemic”, and “Impacts of Mental Health and Wellbeing”. Given the importance of documenting mental health consequences of the pandemic, a dedicated analysis was applied to identify themes and subthemes which most accurately described the impacts of the pandemic upon stakeholder wellbeing specifically. These themes were revised by the research team and reformed until a final set of themes which best represented the data were selected.

## 3. Results

Thematic analysis of the data generated three themes representing the most salient points discussed by interviewees which pointed to the impact of the pandemic upon overall wellbeing. Within each theme, a set of subthemes emerged. These are shown in Table 1.

A summary of themes, subthemes and the units of code contextualising these themes can be found in Appendix A.

### 3.1. Changes in the Way the Industry Communicates

#### 3.1.1. Teleconsultation

One of the most significant disruptions resulting from the pandemic came in the form of changes to communication style. In concordance with the public health guidelines, a reliance on digital and telephone communication alone became especially relevant for equine veterinarians who were required to use voice or video calls (telemedicine) to assess non-emergency cases. Decision making around which conditions veterinarians felt warranted in-person attention was a source of anxiety for some. Although frustration was expressed towards governing bodies of the veterinary profession, there was also empathy shown towards those within the Royal College of Veterinary Surgeons (RCVS) and British Equine Veterinary Association (BEVA) who faced the challenge of producing guidelines that addressed the governmental approach to public health without compromising animal welfare. Clearly, situations in which actions to protect equine health may pose a threat to human health require careful consideration. Veterinarians discussed how, in the initial stages of the pandemic, rapidly changing information and the slow release of a definitive framework for decision making made them feel responsible to decide which treatments warranted in-person attention, and which they should manage through telecommunication. Veterinarians also recounted situations in which they felt their health was put at risk whilst attending yards before the initial lockdown, demonstrating the important role that telemedicine could play in protecting the health of both the public and veterinarians themselves.

“We had quite a lot of difficulty deciding what we should or shouldn’t be doing because it was a situation that no one had faced before. The first challenge I think was saying what we need to stop doing as normal, and that decision came from the vets and not the management, which maybe isn’t, so much, well, everyone was just confused with what they were doing. But in the time running up to lock down, we were still out doing vaccinations and teeth rasps and social distancing wasn’t a thing. We kind of started to realise that we were not really happy doing this when we were meant to be heading toward a lock down.”—V2

In equine first opinion practice, scenarios were often described where social distancing presented a more immediate threat to safety to the vet than contagion, demonstrating the front-line nature of the veterinarian’s role. As one equine veterinarian said of balancing human health risk whilst treating a potentially dangerous horse: “You feel like you would rather take your chance with coronavirus than get crushed or kicked”—V1

Conscious of the social responsibility they felt to minimise the risk of coronavirus transmission within the community, veterinarians also indicated fear around contagion, particularly for their clients. This culminated in genuine anxiety for some, as demonstrated in the following quote: “My fear as a vet is that seeing ten different people in a day, and at times you are in close contact with them whether you want to be or not, I was really worried that if I caught it and didn’t have any symptoms, then the number of people I could have been in contact within a week is crazy!”—V4

Even after 23 March, when the country officially entered a lockdown state, veterinarians encountered equine premises that continued to function as they had before the pandemic. This lack of concern for biosecurity left veterinarians feeling that their safety was often overlooked by horse owners and yard managers.

“You would go to yards in the first three weeks and you’re like, “are you guys aware there is a pandemic?” Everyone and their mother are up at the yard. “Oh great!”.”—V1

Some placed the onus of responsibility for biosecurity upon the yard manager or horse owner, whilst others felt that human biosecurity lagged behind the veterinary industry with regard to infectious disease control. There was also a sense that practicing veterinarians may have liked to contribute to conversations around more global and public health issues, but that their opinions had not necessarily been sought.

“I find that disappointing… the intensive piggery industry, and the hen and chicken industry, where we have strict biosecurity- the farmers have been doing that for decades. I feel that we missed a trick there with care homes, in terms of straight forward biosecurity- with having a simple shower-in-shower-out system- which didn’t happen as far as I understand.”—V2

The switch to telemedicine was welcomed by the veterinarians interviewed for the benefits it offered to public health, but in practice for them personally it was viewed from a mixture of positive and negative perspectives depending upon their personal preference. Figure 1 summarises the positive and negative aspects of telemedicine discussed.

On balance, the positives and negatives of telemedicine for equine veterinary management were largely equal. Figure 1 shows that reduced costs, improved relationships and saved time were benefits of telemedicine. Conversely, being manipulated by clients who exaggerated their horses’ condition to secure yard visits, judging disease severity, and increased time and workload were negative aspects of the technology that faced veterinarians.

All veterinarians were in agreement that face-to-face visits would be preferable. However, opinions were divided as to how useful technology was and whether or not they saw its use continuing post-pandemic. For one veterinarian, telemedicine was an important part of their normal working practice: “I find [telemedicine] so useful, I can’t do without it. I think there is a general fear that, “I’m not going to take my animal to the vet because that is going to cost me £35”, so I have been encouraging WhatsApp for years.”—V2

Financial benefits of telephone communication with their veterinarians may encourage owners to seek advice more readily in the assurance that they are not going to be billed for the service, ultimately leading to improved animal welfare through more timely treatment of time-sensitive ailments. Although others noted that their clients’ perception of phone advice as a free service made the switch to telemedicine more complex as they needed to balance the expectations of horse owners with the business model of their employers.

“The problem is that our company is really keen for us to do phone consults and videos, but we already do a lot of that, and people will often be on the phone for advice two or three times a day. We don’t charge for that so it is kind of hard, because it is the same situation as before lockdown- you feel like you couldn’t really charge for that”—V1

In terms of the practicalities of telemedicine, four out of five veterinarians found that teleconsultations were more complicated and time consuming, with a primary concern for misdiagnosis.

“you can send videos and get pictures to supplement the descriptions, but I think the main thing is that there is a lot more work to that and not enough time in the day…I think there is more room for misdiagnosis, and that’s something I worry about, but it’s so busy. In the phone calls as well, they will start talking about something completely different, going off topic, then the information that you actually want to know, like dates and how long something has been like that for, if they are on any medication, that is the information they don’t know and you can spend a long time on the phone in a situation like that.”—V3

Capitalising on the requirement for veterinarians to visit cases requiring immediate veterinary care, there were accounts of horse owners using telephone calls to manipulate veterinarians into visiting their yard, despite their horses not requiring urgent care.

“the trouble is that people got wise to the fact that we wouldn’t come out to a routine tooth rasp, and they were phoning up the practice maybe saying that they had a horse that was quidding, then you would get to the yard and you think, “Yeah, there is no reason for them to be quidding”.”—V4

Such behaviour eroded the trust of veterinarians toward their client, with one saying: “I think clients, not all, I mean I’ve got loads of brilliant clients that I love to bits, but people are selfish. That is actually what it shows.”—V4

#### 3.1.2. Digital Training

For some owners, adapting to the restrictions involved innovating new ways to engage with trainers through digital means. Some engaged in online training using the video on their phones, which allowed riders to maintain their routines, promote their sense of achievement and provided peace of mind that the horse’s fitness or training would not deteriorate. Training was also an important activity for those who kept their horses at home for preventing boredom and maintaining contact with friends.

“for me, it has been positive for my riding. I have been doing Zoom lessons with my chum who teaches me, so we were doing lessons when we weren’t allowed to do anything”—HH4

There were also opportunities to enter showing and dressage classes online. Although not a novel concept, these online events may have increased in popularity during travel restrictions [16].

“I really enjoy having something to aim for … there have been a few shows online and got my husband to video and send it in. That has kept us going and it gave us something to do.”—HH3

It was not only horse owners who benefited from online training opportunities. Farriers were also able to access online training events and competitions to fulfil professional requirements in the form of continued professional development (CPD). The positive aspects of such events being hosted online included the possibility to store and re-watch recordings: “they have guys they employ to do clinics all over the world and they have been doing those online… the good thing is that it is there, so you can go back and refer back to it if need be. There are now online farrier competitions as well… it is a really good idea.”—F4

#### 3.1.3. Social Media

For the majority of interviewees, the use of social media was not stated as an important factor involved in the care of horses. However, during the pandemic, there were notable advantages and disadvantages for the equestrian stakeholder community in having access to social media.

Social media may have eased and improved information sharing for welfare centres with the responsibility to generate and disseminate information for the equestrian community in the UK and worldwide. Farriers used social media platforms, e.g., WhatsApp and Facebook, to unify their approach and provide support across the local farrier community, as well as to share consistent information with their clients.

There were also corrosive effects of social media upon the wellbeing of the community. Figure 2 shows direct quotes from interviewees expressing the sentiment of these social media attributes and drawbacks.

Social media platforms had several positive uses during the pandemic, which included improving information sharing and promoting community amongst peers and colleagues. However, negative impacts on wellbeing, such as peer–peer judgement and jealousy, were also noted in interviews (Figure 2).

### 3.2. Restricted Contact with Horses

#### 3.2.1. Closure of Yards

Restrictions on premises that prevented or limited owner interaction with their horses was a cause for concern in relation to the impact on the emotional wellbeing of horse owners. Farriers who were interviewed worried for the mental health of their clients, saying that some livery yard owners who closed their doors during the pandemic had “overstepped their remit”—F1 and were depriving owners of valuable contact with their horses “when they needed them the most”—F4. Motivations for enforcing public access restrictions on yards were complex and multifaceted. Decisions to close yards were dependent on the individual circumstances and principles of yard owners but were strongly influenced by livery clientele. A horse owner who also owned a livery yard was forced to re-open their doors to their clients sooner than anticipated due to serious concerns for the mental health of their clients.

“for the first 6 weeks [clients] were [saying]- do what you need to do to keep the staff and yourself safe. Totally trust you to look after the horses. But on week 7 that had slipped to- why can’t you open, other yards are open, some yards didn’t close, can’t you look at them and find a way through this? I realised that for 2 of the ladies, this was becoming a mental health issue. So, at that point I decided to get the clients’ back, but on a timetable where they had little to no contact with the staff.”—HL2

Re-opening doors was also not without consequence for the safety and wellbeing of staff and professional visitors to the yard, who may have felt some security in the knowledge that their employer had taken action to limit footfall. Such considerations were an important part of decision making for yard owners and employers.

“I am responsible for the health and safety of my staff, which includes safety from coronavirus, but also mental health. But I am also responsible for the safety and well-being of the horses and the client. At the end of the day, as I explained in a staff meeting, without the clients—who could at any time come and pack up their horse and go elsewhere- without them there would be no income, and there would be no money to pay staff wages. So, reality has to kick in and you need to find a way through this.”—HL2

The actions of other professionals were at times dictated more by the concerns of their clients than by their own. Some farriers felt that they were at greater risk of contagion “going to Tesco”—F1 than visiting quiet yards in rural locations. Several of the biosecurity practices they employed were used to reduce anxiety in owners, rather than from their own perception of risk.

“It would be other people’s worries that would dictate the precautions we put in place”—F1

#### 3.2.2. Concern for Horse Health

In their absence from the yard, some horse owners feared for their animal’s health during restricted yard visits. Detailed accounts of the reasons for this concern with regard to equine health and wellbeing are detailed elsewhere [9]. In regard to the repercussions for human wellbeing, pandemic-associated obstacles to horse care were frequent drivers of distress and anxiety in horse owners. One farrier described a phone call from a horse owner in “floods of tears because her farrier had refused to go up because no one was allowed on the yard to put a shoe back on her horse”—F4. This quote demonstrates the highly emotive nature of a horse’s health and wellbeing for their owners, a concept which is true even when the professionals overseeing the care for the horse are not concerned. For example, farriers may have continued trimming horses’ feet even if the horse was not a priority visit in order to quell concerns of owners.

“There were some animals that needed done, but the trims especially, if we were doing them every 6 weeks, it was for the owners’ benefit”—F1

Similarly, owners were incredibly concerned around vaccinations lapsing, but veterinarians were instructed by governing bodies that routine vaccinations were not essential visits during lockdown. In a severe example, worry over their horses’ vaccination programme led an owner to pressurise their veterinary practice into providing vaccinations.

“I had to ask a number of times- I need you to come out and do vaccinations, you’re coming out anyway. I was able to work with them eventually, through coordinating with the other horses that needed to be seen. Then by that time, he needed his teeth done, he was overdue, and the poor vets eventually said we will just do it, to save arguing with me really”—HL1

#### 3.2.3. Loss of Positive Interactions

It was not only a case of being concerned for their horse’s wellbeing that led to increased anxiety in owners. A lack of control over their horse’s routine, reduced physical contact with the horse, and loss of interaction with peers on yards or through equine events were important factors potentially contributing to reduced horse owner wellbeing. Those who maintained unrestricted access to their horses described their experience during the pandemic as largely positive. All owners at home, and those who had been able to continue seeing their horses as normal, referred to being fortunate to be in such a position. Those who had experienced the impact of the pandemic in other areas of their lives, such as furlough, emphasised the importance of their horses as a source of emotional support at that time.

“With everything going up in the air, work and all that kind of stuff, I mean, I wouldn’t have coped at all if my horses weren’t at home and I couldn’t see them, it would have been, would have been a heck of a lot worse. Yeah. It would have been awful, actually.”—HH5

Owners tended to reference their structured routines, physical activity, engagement with nature and the outdoors, and the companionship offered by their horses as sources of contentment during the pandemic and empathised strongly with those who had lost this. However, restricted access to their horses had some notable benefits for some owners. Limitations on the time they could spend with their horses involved having a time slot within which to attend, and this allowed them to organise their life outside of equestrianism more effectively. One livery yard-based horse owner said of their peers: “A couple of folks said that they weren’t telling their work they were off the two-hour rota [system] because it suited them so well! It meant that they had that time sectioned off to go and have their horsey time!”—HL4

As restrictions were eased, owners valued their time interacting with their horses in ways other than riding, gaining the benefits of simple interactions with their animals.

“Not necessarily more riding, I was just hanging out there, brushing them, pulling off ticks. All the things that you never really had time to do.”—HL3

The closure of competition and event spaces also gave owners an opportunity to focus on their horse’s development in ways that they did not expect. For example, one owner interviewed was able to spend more time with their young horse: “I didn’t think I would have time this year to back my Clydesdale, because I thought it would be too busy eventing and working … every cloud, you know?”—HL5

### 3.3. Pro-Social Behaviour

#### 3.3.1. Virtual Socialising and Fund Raising

The equestrian community interviewed provided several examples of altruism, pro-social action and community engaging behaviours. Those who discussed being involved in these activities tended to speak more positively about how the pandemic had impacted their own lives, perhaps having been positively affected by the sense of community garnered from taking action to help others. One livery yard took action to address the loss of social opportunities on the yard due to social distancing and time slot rules. Together with their livery yard manager, this group arranged virtual quiz nights, online training sessions and other online activities to provide a space for friends to meet and interact as they usually would have. In addition, the organisers added a small charge for joining the events in order to raise money which was donated to charities set up to aid equestrian business during the pandemic.

“… smaller yards would be struggling a lot more than we were. So, we have been, not only looking after us, but also raising money for other yards as well. [The yard manager] did Zoom pub quiz nights so that we could all keep in touch with each other- because we are all really sociable and we are all friends. So, because we couldn’t have nights out, we had virtual nights out on zoom, and just paid a tenner or something to do it. Then all the money went toward to BHS fund”—HL5

#### 3.3.2. Supporting the Community

Another yard manager, in noticing how much their livery clients were missing out on interaction with their horses, took weekly pictures and videos of each individual horse and shared these with the owners directly.

“every Tuesday we pulled out all of the horses and took pictures from every angle and that was our condition score day. So, our clients could see their horses and know how their condition was…they wanted something on their phone, a cute photo or a little video or the condition scores. It made everyone feel that they were still part of the yard and that is what it should be. Everybody should be happy on a yard and enjoy their time. Particularly the horses.”—HL3

By providing personalised interaction, albeit through digital means, this yard owner sought to nurture the horse owners’ need for interaction with their horse, as well as connection with peers.

Farriers also demonstrated a strong sense of solidarity amongst themselves, noting that they are a “small community and we all try to help each other”—F4. This was evident the fact that all farriers interviewed had taken on clients for friends and colleagues who were shielding, but all emphasised the importance of directing those owners back towards their original farrier once the pandemic was over. Towards their own clients, farriers showed a great deal of empathy, often having built strong relationships with them over many years of regular visits. One farrier even provided shoes and trims for a large yard of horses for free in the spirit of community support: “we have just shod them and not charged for this cycle… it is about us all working together to minimise the impact”—F1

#### 3.3.3. Valuing Horses

Finally, interviewees generally expressed gratitude that they had their horses, their employment or their yard friends to help them overcome challenges that presented during lockdown and as society moved towards less stringent restrictions. Equine welfare centre charity managers summarised this by highlighting how important it is for horse owners to take stock of the aspects of horse care that promote their happiness, hopeful that individuals who own and care for horses were able to take something positive from the pandemic.

“people really value the fact that they do have a reason to go out in the morning and poo pick. The stuff that we may be used to see as a bit of a bind or an inconvenience, suddenly becomes the highlight of the day due to the pandemic. And I hope that people don’t lose sight of that.”—WCM4

Reflecting upon what they had learned from the pandemic, and what practices they would try to bring forward with them, a welfare centre manager said: “I think just taking the time to enjoy the journey with the horses. You know, just taking that time to enjoy it.”—WCM2

## 4. Discussion

The findings of the present study allow the exploration of the many facets of human life associated with the care and welfare of horses which were impacted by the COVID-19 pandemic. For a subpopulation of equine industry stakeholders, these changes in routine and lifestyle had both negative and positive consequences for overall wellbeing. Although the findings represent the perspectives of a small cohort, this study may have applicability for the wider community through presenting the nuances of human behaviour which can contribute to research on human–animal interactions and the benefits for wellbeing that such relationships can have during times of stress.

The increased reliance upon digital communication to manage animal care and welfare helped stakeholders to conform to public health guidelines but may have negatively impacted human wellbeing. For some interviewed veterinarians, a significant change in working practice to incorporate telemedicine precipitated anxiety over deciding which cases should receive visits and which could be managed through teleconsultation. Decision making regarding the provision of appropriate care for animals has been, and continues to be, a potential source of stress for veterinarians [17]. During and after the pandemic, veterinarians had additional considerations to take into account when assessing cases, and a recent survey of 540 veterinarians revealed that 46% found decision making around service provision and visits “ethically challenging” during the COVID-19 pandemic [18]. It is possible that pre-existing stressors for veterinarians may have been exacerbated by the accumulation of challenges that presented in day-to-day practice as well as those associated with the pandemic, including teleconsultation.

The results from the present study conform with that of Butler et al. [15], who noted that the majority of an equine veterinarian sample found teleconsultation to be excessively time consuming, preferring to physically assess cases, although, some found that consultations took less time; consistent with findings of Bishop et al. [19]. A possible contributor to strained communication over telephone could be the loss of subtle interactions which normally punctuate person to person communication. Previous research has suggested that non-verbal communication may be particularly important in veterinarian–client relationships [20], and this would often have been lost during teleconsultation.

A particular challenge to the veterinarian–client relationship was the lack of trust shown by horse owners toward their veterinarians. This was displayed through owners’ attempts to persuade veterinarians to perform vaccinations, routine dental check-ups, and performance-associated assessments, which were deemed by the profession as non-essential during the UKs emergency lockdown phase of the pandemic. Given the front-line position that veterinarians held, it is important to highlight the burden of responsibility that many felt to protect human health, and for this reason decisions around animal welfare during this time were made with great consideration. In fact, involving veterinarians in COVID-19-related policy development at the local and governmental levels has been discussed as a promising strategy to improve the pandemic response, feeding into the One Health concept for better public health outcomes [21,22]. However, many non-veterinary equestrians grew frustrated and concerned for their horses’ health as vaccination programmes were extended or lapsed due to veterinary practices’ decision to restrict all non-essential visits [11]. Clients acting out these frustrations by targeting veterinarians with manipulative or confrontational behaviour are likely to have had detrimental effects upon veterinary practice staff and veterinarians alike, who would then be forced to defend their profession’s position on matters of equine health and welfare. Client mistrust resulted in a negative impact upon the mental wellbeing of veterinarians, and effects of a lack of solidarity from clientele are aptly described by Sumner and Kinsella [23]. This commentary explains how a lack of empathy and respect from those they are working to protect may lead front line workers to develop occupational stress and burnout, conditions with a pre-existing connection to the veterinary profession.

Heavy physical and emotional workloads have long been associated with reduced mental health parameters in the veterinary profession [14,17,24,25,26,27]. Even before the pandemic, the prevalence of suicidal ideation, depression and anxiety recorded in veterinarians [27] was cause for concern from veterinary governing bodies such as the Royal College of Veterinary Surgeons (RCVS) and the American Veterinary Medical Association (AVMA). As a profession at risk of poor mental health, the additional challenges posed by the pandemic will have compounded the risk for equine veterinarians and nurses [28]. Initiatives supporting mental health and wellbeing in the veterinary profession, such as the RCVS’ Mind Matters Initiative [29] and the AVMA’s Wellbeing and Peer Assistance Initiative [30] act to fulfil the positive interventions for mental health and wellbeing outlined by researchers of psychological wellbeing in the veterinary profession [31]. Mediation to promote better veterinary mental wellbeing throughout the careers of veterinary professionals could involve integrating resilience training into the veterinary curriculum, improving availability of information about mental health, and providing accessible counselling resources. These interventions may be especially valuable to the profession as they emerges from the pandemic having overcome, what may be, one of the most challenging time periods they have faced.

Loneliness, isolation and low mood were predicted outcomes of social distancing and “stay at home” messaging [2] and the detrimental effects of the pandemic on mental wellbeing have been widely documented using survey-based methods [1]. Physical activity largely increased across the UK [32]. However, the exercise that individuals engaged in may have been a temporary surrogate for their preferred hobby, meaning that the full mental health benefits of the activity were not realised. Fullana et al. [4] noted that pursuing hobbies and maintaining a sense of routine were two ways in which depressive symptoms could be counteracted during the pandemic. For some horse owners, this was achieved by turning to digital training and competition with their horses. Reports of the value of digital devices as a way to maintain connection in a landscape of isolation have shown that embracing new means of communication may provide significant benefits to wellbeing [33].

For a large proportion of society, social media plays a key role in the relationship between individuals and their devices. The present study showed that there were equestrian industry stakeholders who found information sharing to be simplified by social media and this reinforces findings from literature which highlight its use to support accurate information dissemination during the pandemic [34]. However, depending upon its use, social media use may have implications for psychological wellbeing, with judgmental and negative commenting having negative effects upon the emotional and psychological wellbeing of those receiving the messaging, as well as the sender [35]. Results from this study showed that engaging in or being exposed to negative COVID-19-related discussion and social media judgement had a negative relationship with life satisfaction, and that measures of life-satisfaction were reduced in those survey respondents who felt that social media was a suitable platform to express inappropriate opinions. As such, social media may play a role in both protecting against, and exacerbating mental health issues, but this appears to be dependent upon its use and the personal attributes of the user. It may be that those with restricted access to their horses would be more vulnerable to negative emotions such as jealousy and judgement when viewing content from un-restricted horse owners, leading to the feeling of “unfairness” around their circumstances.

Unlike companion animals kept at home, horses are often stabled at livery yards located separately from the owners’ home, where aspects of day-to-day care may be delegated to employees. Owing to their status as business establishments, many livery yard owners made the decision to prevent non-essential workers from accessing the yard, including horse owners. Several studies have described disruption to horse and owner routine resulting from the closure of livery yards, as well as the negative consequences for the mental wellbeing of the owners that such restrictions may have had [9,10,11,13]. The decision by yard owners to close was often driven by the careful balancing of responsibilities toward the horses and humans who came into contact with the yard, as has been extensively documented by Furtado et al. [13]. The decision to re-open required consideration of the psychological consequences that a lack of contact with their animals may have upon horse owners. Participants in this study noted, or witnessed first-hand in others, the distress which resulted from restricted access to horses.

Physical interaction with companion animals may help to relieve symptoms of stress and anxiety, promote physical and mental wellbeing in owners and offer a valuable source of companionship during the pandemic [36,37]. One study which looked at the influence of lockdown upon human–animal relationships revealed that 94% of horse-owning respondents felt that their horses helped them to cope and kept them active [12]. Reliance of owners upon their animals as a source of psychological and emotional support was highlighted by several stakeholders in the present study. These effects may also be due to way that animals facilitate natural human–human connections to form around a common interest in animals, known as a social bridging or buffering effect [12,37]. This element of the human–horse bond may have played a strong role in the negative effects of yard closures on horse owners, where they were unable to access their animal for support, as well as being disconnected from the social network they had formed through that animal. A survey of cat and dog owners found that 49.2% of respondents experienced “a lot”, or “quite a lot” of negative effects on their lifestyle caused by isolation [37]. Such findings support the theory that the lifestyle that a pet affords an owner is an important contributor to the wellbeing benefits derived from ownership [38].

Researchers have suggested that the strength and type of relationship that exists between a human and their animal may actually predict psychological vulnerability in owners [12]. The present study noted that farriers and owners with horses at home were particularly worried about horse owners whose access to their horse had been restricted. Some of these comments were gender focused, and studies have shown that females are more likely to experience psychological consequences from the pandemic [5,33]. With regard to their animals, female respondents were also 1.72-fold more likely to be in the group that gained the most support from their companion animal [12]. It may, therefore, be important to note that the horse owner group interviewed for this study were predominantly female and as such, results should be interpreted with an awareness of the increased risk of negative mental health effects that this sample may have been exposed to.

Evidence has shown that taking part in pro-social activities could positively influence mental health and wellbeing [39]. The present study showed that social initiatives and charitable acts were undertaken by stakeholders, and an overall sense of gratitude was expressed at being able to continue caring for horses during the pandemic. Interviewees who discussed taking part in activities offering a sense of purpose, community and social belonging were distinctly more positive in their overall discussions around the pandemic and their overall satisfaction at the time, based upon the reduced number of negative terms and intonations expressed in interviews. The potential changes in horse owner perception following pandemic-associated isolation were explored by Hockenhull and Furtado [40], who suggested that the pandemic may have offered equestrian stakeholders an opportunity for reflection upon how they derived their satisfaction from involvement with horses. Equestrians deriving pleasure from undertaking tasks other than riding, such as grooming and general horse care, could be a positive outcome of the pandemic that others have also noted [41], and this phenomenon may have positive impacts on owners’ relationships with their animals in the future.

The primary limitation of this study is that it provides only a rudimentary insight into the overall impact of the pandemic upon the stakeholder group, and qualitative investigation with a focus on wellbeing is warranted in order to fully explore the lasting impacts of the COVID-19 pandemic on mental health and wellbeing. Furthermore, results from this study may not accurately represent the experiences of communities outside of the stakeholder group studied, meaning that the applicability of the findings for wider groups is limited. However, the value of this investigation lies in the more particular observations of human nature and responses to emergency scenarios, which might inform researchers, veterinarians and equestrian industry stakeholders of the challenges the equestrian community has faced.

## 5. Conclusions

This study provides a unique insight into the ways that the COVID-19 pandemic impacted a distinct group of equestrian industry stakeholders who experienced a variety of positive and negative consequences upon mental health and wellbeing. Social isolation, changes in communication and restricted access to horses caused social, professional and psychological stress in the group of interviewed individuals. These impacts on wellbeing may be relevant for consideration in future emergency scenarios when developing strategies to protect mental health across the equestrian industry. Equine veterinarians highlighted the psychological impacts of challenging clients and unrealistic workloads- symptoms of the pandemic which could be countered by promoting awareness of these potential mental health concerns being amplified during global emergency scenarios. The closure of livery yards to horse-owning clients was shown to have provoked industry concern for the negative consequences that this can have for horse owners’ wellbeing, results which highlight the key role of the horse in many horse owners’ social and emotional support network. Engagement in pro-social and altruistic activities that benefit the equestrian community positively influenced mental health, wellbeing and resilience of interviewees. Such findings add credence to the suggestion that promoting a sense of community and co-operation could be protective against stressors arising from the pandemic.

## Figures and Tables

**Figure 1 animals-11-02163-f001:**
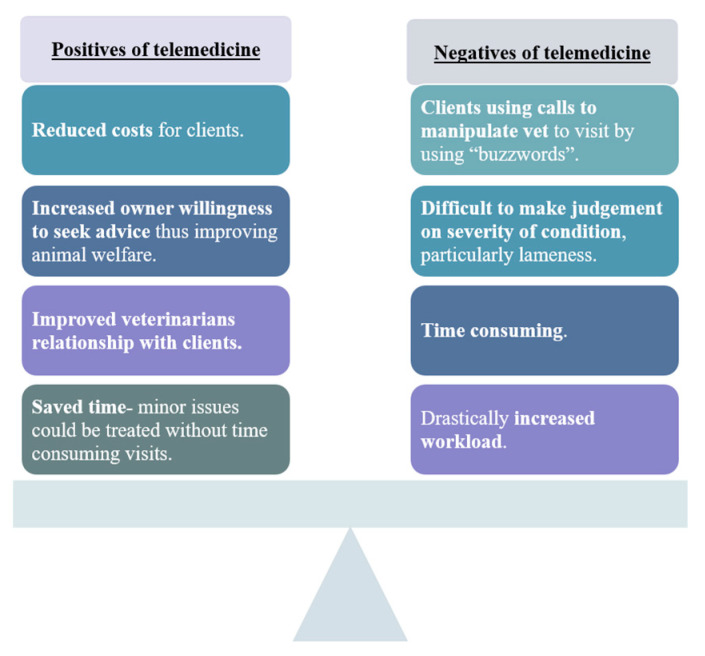
The positives and negatives of telemedicine for equine veterinary management.

**Figure 2 animals-11-02163-f002:**
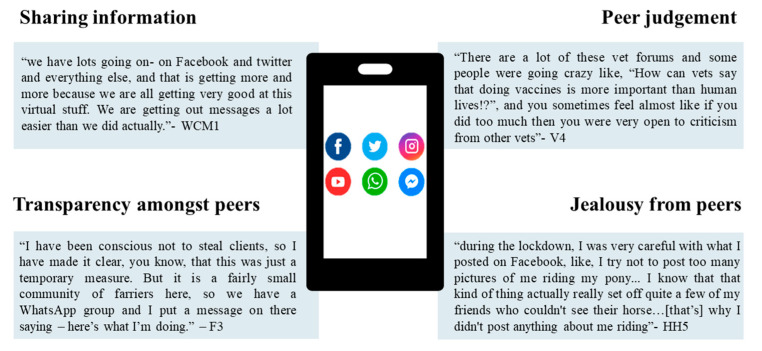
Negative and positive aspects of social media use within the equestrian, farrier, and veterinary community during the first phase of lockdown.

**Table 1 animals-11-02163-t001:** Themes and subthemes relating to mental health and wellbeing as discussed by equestrian stakeholder interviewees.

Theme	Subthemes
1. Changes in the way the industry communicates	Teleconsultation Digital training Social media
2. Restricted contact with horses	Closure of livery yards Concern for horse health Loss of positive interactions
3. Pro-social behaviour	Virtual socialising and fundraising Supporting the community Valuing horses

The themes and subthemes presented were identified after noting the prevalence of negative influences upon wellbeing that interviewees discussed as they described the impact of the pandemic upon their management of native breed ponies.

## Data Availability

The data collected for the present study may contain details which could potentially lead to the identification of study participants. As such, requests for access to this data may be sent to SRUC’s Data Protection Officer. Address: Scotland’s Rural College, Executive Office, Peter Wilson Building, Kings Buildings, West Mains Road, Edinburgh EH9 3JG. Telephone: 0131-535-4432. E-mail: dpo@sruc.ac.u.

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
