# Peer review of "The Human Aspect of Horse Care: How the COVID-19 Pandemic Impacted the Wellbeing of Equestrian Industry Stakeholders"

_animals, 2021, doi:10.3390/ani11082163_

Round 1

Reviewer 1 Report

A valuable research study that overlaps the interesting and critically important area of horse and human wellbeing.   Generally well written with a few instances of slightly unclear wording/phrasing.  My only negative comment is that more detail is needed in the method section.

Detailed comments are provided on the annotated manuscript provided. 

Reviewer 2 Report

Animals-1299009-peer-review-v1

The article has relevance in view of the global scenario. However, my main point is related to the low number of interviews (n=24). The n is too low for inferences to be made for the entire equestrian community. I suggest a review of the discussion and conclusion.

The paper is well written. However, I have some suggestions as to the structure and organization of the sessions.

Keywords:

Review keywords because the title words cannot be repeated

Reorganize:

Line 69 to 74 – Include in session 2 Materials and methods

Line 74 to 77 – Include in session 3 Discussion

Line 78 – materials and methods

Include what the meaning of: V, HH, HL, etc.

Example: veterinarians (V), farriers (F)…

Line 97 – What means firu 1?

Line 557 Conclusion – rewrite based on the objective

This study provides a detailed overview of the impact of the pandemic upon distinct cohorts within the equestrian industry. It has shown that responses to the pandemic in the veterinary and equestrian industries have resulted in negative consequences for the wellbeing of individuals, but that there have also been several positive outcomes for mental health and wellbeing. Social isolation, changes in communication and restricted access to horses produced social, professional and psychological stressors which should be considered in future emergency scenarios. Decisions should balance the need to protect mental health and wellbeing of humans as well as the welfare of the resident horses. Partaking in pro-social and altruistic activities that benefit the equestrian community can have a positive influence on mental health, wellbeing and resilience against stressors arising from the pandemic.

Line 564 to 571 – Include in the discussion

Protecting equine veterinarians from unmanageable workloads and promoting a unified approach toward the pandemic within the profession may help to reduce stress related factors that veterinarians may experience during national or local lockdown requirements, especially as they juggle conflicting priorities between human and equine health. Proprietors of livery yards may be forced to close their doors to horse owning clients owing to governmental guidelines, but the negative consequences that this can have for horse owners should be carefully considered in decision making around access.

Line 575 to 579 – Include in the discussion

The limitations of the study are associated with the limited applicability of the findings for wider communities. However, the value of this investigation lies in the more particular observations of human nature and responses to emergency scenarios, which might inform researchers, veterinarians and equestrian industry stakeholders of the challenges the equestrian community has faced.
